# Sleep Disturbance in Chinese College Students with Mental Health Problems: A Moderated Mediation Model

**DOI:** 10.3390/ijerph192114570

**Published:** 2022-11-06

**Authors:** Yanping Sun, Lin Wang, Chang Li, Wanshu Luo

**Affiliations:** 1Department of Applied Psychology, College of Sports and Health, Shandong Sport University, Jinan 250102, China; 2Student Affairs Office, Shandong Sport University, Jinan 250102, China; 3Department of Insurance, Shandong University of Finance and Economics, Jinan 250014, China; 4School of Dance, Shandong Youth University of Political Science, Jinan 250103, China

**Keywords:** hostility, somatic symptoms, obsessive-compulsive symptoms, sleep disturbance, college students with mental health problems

## Abstract

Sleep disturbance has an enormous impact on college students. Poor sleep is associated with low academic achievement, psychological distress and high health risk behaviors. College students with various mental health problems (e.g., anxiety and depression) are particularly at risk for sleep problems. The aim of the present study was to examine the impact of a wide range of internalizing/externalizing psychological problems on sleep disturbance. A total of 2134 Chinese college students (60.2% men) with mental health problems were selected as participants after completing the self-reported Chinese college student mental health screening scale. A web-based survey was used to assess a wide variety of internalizing/externalizing psychological problems and sleep disturbance. The results showed that hostility, somatic symptoms and obsessive-compulsive symptoms (OCS) were significantly associated with sleep disturbance. Somatic symptoms played a mediating role in the relationship between hostility and sleep disturbance. Moreover, the mediating effect was moderated by OCS, and a significant difference in the mediating effects was observed between low OCS and high OCS groups. Overall, our research findings indicate that a high level of OCS exacerbates the adverse effects of somatic symptoms on sleep disturbance, and suggests that assessment and improvement of hostility, somatic symptoms and OCS should be considered in facilitating better sleep among college students with mental health problems.

## 1. Introduction

Sleep is an essential physiological and psychological process for our mental health and well-being [1], and effective sleep plays a vital role throughout life in the development of our physical, cognitive and psychosocial functioning [2,3,4]. On the other side, sleep disturbance, which refers to persistent difficulties falling asleep, frequent awakenings and inability to resume sleep [5], predicts low academic achievement [6], depression and/or anxiety [7], poor peer and/or intimate relationships [8] and high health risk behaviors such as smoking, drinking and use of stimulants [9]. Notably, the Annual Sleep Report of China 2022, issued by the Chinese Academy of Social Sciences on World Sleep Day, reported that more than 90 percent of Chinese college students had average (49%) and poor (45%) sleep quality, and nearly half of the college students went to bed after 12 a.m. A considerable portion of college students reported a lack of sleep caused by sleep delay [10]. Furthermore, Lund and her colleagues [11] found that about 60% of American college students have sleep disorders, and 17.9% use hypnotic drugs to improve their sleep. 

Given the high prevalence of sleep disturbance and its detrimental effects, sleep disturbance has gradually become a notable public health concern. Thus, it is crucial to explore (a) factors that are associated with sleep disturbance, and (b) mechanisms influencing sleep disturbance. This study aims to offer a theoretical foundation and empirical support for the prevention and control of sleep disturbance in college students. Sleep disturbance is intricately associated with internalizing psychological problems (e.g., depression, anxiety and somatic symptoms) [12,13] and externalizing psychological problems (e.g., hostility, internet addiction disorder and obsessive-compulsive symptoms) [14,15,16]. Internalizing psychological problems refers to negative emotions experienced by individuals, and externalizing psychological problems refers to externally directed problems such as aggression and hostility [17]. Previous studies have reported that somatic symptoms, hostility and obsessive-compulsive symptoms (OCS) are risk predictors of sleep disturbance [18,19,20,21]. For example, somatic symptoms/complaints were significantly correlated with insomnia/poor sleep quality in samples of adolescents [22], postmenopausal women [23], people with traumatic brain injury [24] and Japanese adults [25]. Correspondingly, Schlarb et al. suggested that somatic complaints were also closely correlated with poor sleep quality in university students, and depression moderated the relationship between somatic complaints and sleep quality [26]. Nordin et al. indicated that somatic symptoms, including stomach pain, back pain, nausea/gas/indigestion and dizziness were closely correlated with sleep disturbance [13].

In addition to somatic symptoms, externalizing psychological problems, including hostility and OCS, have also been associated with sleep disturbance [20,21]. For instance, previous research has shown that hostility was a significant risk factor for sleep disturbance in normal sleepers [20] or an employee population [27]. Similarly, Tsuchiyama et al. suggested that subjective sleep measured by the Pittsburgh Sleep Quality Index was predicted significantly by hostility evaluated by the Cook–Medley Hostility Scale in healthy adults [14]. Furthermore, Hyphantis et al. showed that in comparison with both the healthy participants and medical patients with lower scores on hostility (aggression-hostility, delusional hostility and acting out hostility), significantly more severe somatic symptoms were found in those with higher scores on hostility [28]. Thus, apart from its direct effects on sleep disturbance, hostility possibly influences sleep disturbance indirectly with somatic symptoms as a mediator. That is, hostility has an impact on somatic symptoms, which in turn disturb sleep.

Obsessive-compulsive symptoms are an independent risk factor predicting sleep disturbance. For instance, Paterson et al. found that sleep disturbance is consistently linked with the severity of obsessive-compulsive disorder (OCD) [29]. Moreover, Timpano et al. found a significant association between obsessions and insomnia, and particularly intrusive thoughts have an independent effect on insomnia symptoms [21]. 

Although previous work has shown relationships between internalizing/externalizing problems and sleep disturbance, most of the prior research has only examined how one or two aspects of internalizing/externalizing problems are associated with sleep disturbance. The question remains as to how and when various internalizing/externalizing problems predict sleep disturbance. These combination questions are more interesting and challenging, and are worthy of in-depth study. Thus, with the aim of understanding the mechanism of sleep disturbance comprehensively, we explore the impact of a wide range of internalizing/externalizing problems on sleep disturbance. Moreover, our study focuses on college students with mental health problems (e.g., anxiety and depression), for these students are the most at-risk group for sleep problems. 

Specifically, our study explored the impact of a wide range of internalizing/externalizing psychological problems on sleep disturbance in a sample of Chinese college students with mental health problems. The first objective was to identify internalizing/externalizing problems that are particularly correlated with sleep disturbance. A second objective was to examine how and when these crucial internalizing/externalizing problems exert impacts on sleep disturbance using a moderated mediation model as shown in Figure 1. The moderated mediation model analyses can handle issues such as how a predictor variable impacts an outcome variable and when a variable mostly strongly predicts an outcome variable. Thus, we adopt the mediation model analyses to evaluate quantitatively the extent to which a predictor variable may affect an outcome variable through a potential mechanism [30], and whether the indirect effects of the predictor variable on the outcome variable are altered in high/low levels of a moderator. First, we hypothesized that internalizing problems (somatic symptoms) and externalizing problems (hostility, OCS) are significantly associated with sleep disturbance in Chinese college students with mental health problems. Our second hypothesis was that somatic symptoms would play a mediator role in associations between hostility and sleep disturbance, as shown in Figure 1 (hostility 
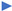
 somatic symptoms 
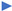
 sleep disturbance). In other words, hostility has indirect effects on sleep disturbance through somatic symptoms. Third, studies about the interaction of hostility, somatic symptoms and OCS on sleep disturbance are very scarce. Though the literature is scanty, we made a bold hypothesis that OCS may moderate the mediating path of somatic symptoms on sleep disturbance, as shown in Figure 1. The moderator variable OCS is a variable that affects the strength of the relationship between somatic symptoms and sleep disturbance. That is, the mediating effect of somatic symptoms on sleep disturbance depends on the levels of the moderator OCS. Thus, the third hypothesis was that the effects of somatic symptoms on sleep disturbance will significantly differ between high levels of OCS and low levels of OCS. Fourth, we hypothesized that the mediating effect of somatic symptoms on sleep disturbance is significantly higher in high levels of OCS than in low levels of OCS. In high levels of OCS, somatic symptoms will exert more negative impacts on sleep disturbance.

## 2. Materials and Methods

### 2.1. Participants

A total of 9131 college students completed the survey in October of 2018 (2298), 2019 (2335), 2020 (2307) and 2021 (2191). They were all first-year students (M_age_ = 18.5 years, SD_age_ = 1.40) from a large university in Shandong province in China. The study protocol was approved by the Research Ethics Committee of Shandong Sport University in China, and conducted in compliance with institutional regulations and guidelines. Informed consent was obtained from all participants prior to the self-reported survey. Questionnaires completed under 100 s were first excluded, then we screened students with mental health problems according to the screening criteria as shown in Table 1. College students were chosen as participants when they met any of the following criteria: Z_I_ > 1, Z_II_ > 2, Z_III_ > 3 or Z_sd_ > 2. Finally, a total of 2134 college students with mental health problems participated in the study. 

### 2.2. Measures

#### 2.2.1. Self-Designed Sociodemographic Questionnaire

Participants provided information on basic sociodemographic characteristics, including gender, only child or not and hometown as shown in Table 2. We classified participants into two groups: one group before COVID-19 (students in this group completed the survey in October of 2018 and 2019), and the other group during COVID-19 (Students in this group completed the survey in October of 2020 and 2021).

#### 2.2.2. Chinese College Students’ Mental Health Screening Scale (CCSMHSS)

This scale is a self-report survey that was developed to assess severe mental problems (hallucinations and delusions and Suicidal thoughts), internalizing problems (e.g., anxiety, depression, somatic symptoms), externalizing problems (e.g., hostility, obsessive-compulsive symptoms, internet addiction) and developmental disturbance problems (school adaptation, academic stress and difficulties in interpersonal or intimate relationships) [31]. As shown in Table 1, internalizing problems include seven indicators, and externalizing problems include seven indicators. The scale contains 87 items with each item rated on a 4-point scale (1 = it is not like me at all; 4 = it is like me very much), and can be assessed as a continuous variable from 4 to 16 (for 4 items) or from 5 to 20 (for 5 items), with higher scores representing more severe mental health problems. The scale has good reliability and validity in Chinese college students, as shown in Table 1.

Sleep disturbance was assessed using the sleep disturbance subscale of the Chinese college student mental health screening scale. The subscale contains 4 items and each item ranges from 1 (1 = it is not like me at all) to 4 (4 = it is very much like me). Total scores ranged from 4 to 16, with higher scores representing more severe sleep disturbance. Participants whose standard score on sleep disturbance is above 3 are classified as the sleep disturbance group; the other participants were considered the no sleep disturbance group. Cronbach’s α of the sleep disturbance subscale was 0.72.

### 2.3. Statistical Analyses

All analyses were conducted using IBM SPSS Statistics Version 23.0(International Business Machines Corporation, Armonk, NY, USA), SAS9.4 (SAS Institute incorporation, Cary, NC, USA) and Mplus Version 7.4 (Mplus Beijing Tianyan Rongzhi Software Co., Ltd., Beijing, China). Statistical analysis methods included descriptive analyses, permutation tests, correlations, logistic regression analyses and path analyses. First, descriptive analyses were performed to obtain the prevalence of sleep disturbance stratified by different sociodemographic characteristics of college students with mental health problems. Sociodemographic factors (e.g., gender, before or during COVID-19) difference were examined by sleep disturbance (standard score on sleep disturbance indicator above 3 was classified as sleep disturbance group) using a permutation test. Second, Pearson linear correlation was conducted to examine correlations between internalizing/externalizing problems and sleep disturbance. Third, multivariable logistic regression analyses were used to explore the specific associations between internalizing/externalizing problems and sleep disturbance. We did not adjust for gender and other sociodemographic variables, because none of these sociodemographic variables were significantly correlated with sleep disturbance. Fourth, we used path analysis to test the theoretical model of the relationship between hostility, somatic symptoms, obsessive-compulsive symptoms (OCS) and sleep disturbance. We established a moderated mediation model, in which somatic symptoms are the mediator for the relationship between hostility and sleep disturbance, and OCS acts as a moderator between somatic symptoms, hostility and sleep disturbance. The path analysis was conducted using Mplus Version 7.4. In the model analysis syntax, bootstrap = 5000, estimator = ML, and type = GENERAL. The Mplus model constraint procedure was used to test whether the mediating effects of somatic symptoms and moderating effects of OCS are significant The multicollinearity diagnostics for the regression analyses were within acceptable limits (tolerance = 0.27 to 0.73; variance inflation factor (VIF) = 1.37 to 3.69). The data indicated an absence of severe multicollinearity. 

Note. *** *p* < 0.001.

## 3. Results

### 3.1. Demographic Characteristics

Table 2 shows the demographic characteristics of the sleep disturbance group (*n* = 78) and the total study sample (*n* = 2134). The sleep disturbance group received standard scores greater than 3 on the sleep disturbance factor. The detection rate of sleep disturbance is 3.66%. The sociodemographic characteristics of the participants with and without sleep disturbance are described in Table 2. For all sociodemographic variables, permutation test results showed that no significant differences were found between the sleep disturbance group and the no disturbance group.

### 3.2. Correlation of Key Internalizing Problem Factors, Externalizing Problem Factors and Sleep Disturbance

Pearson correlation analysis results showed that scores on sleep disturbance significantly correlated with all internalizing/externalizing problem factor scores (*p* < 0.001). Specifically, the total score of sleep disturbance was positively correlated with hostility, somatic symptoms, obsessive-compulsive symptoms and internet addiction disorder (r = 0.327, r = 0.394, r = 0.460, r = 0.363, *p* < 0.001). Somatic symptoms were positively correlated with hostility, OCS and internet addiction disorder (r = 0.550, r = 0.269, r = 0.167, *p* < 0.001). There was also a positive correlation between hostility and OCS (r = 0.354, *p* < 0.001).

### 3.3. Association between Internalizing/Externalizing Problems Indicators and Sleep Disturbance 

Associations between hostility, somatic symptoms, OCS and sleep disturbance are described in Table 3. The multivariate binary logistic regression analysis revealed that hostility (odds ratio = 1.20), somatic symptoms (odds ratio = 1.31) and OCS (odds ratio = 1.32) were significantly positively associated with the occurrence of sleep disturbance. The odds ratios of hostility, somatic symptoms and OCS are greater than 1, which indicates that hostility, somatic symptoms and OCS are risk factors for sleep disturbance.

### 3.4. Mediating Effect of Somatic Symptoms on the Relationship between Hostility and Sleep Disturbance

We constructed a mediating effect model of somatic symptoms on the basis of our theoretical hypotheses. Then, Mplus was used to examine the mediating effect of somatic symptoms on the relationship between hostility and sleep disturbance. As shown in Table 4, hostility positively predicted somatic symptoms (β = 0.55, t = 25.88, *p* <0.001) and sleep disturbance (β = 0.16, t = 5.85, *p* <0.001). Somatic symptoms also positively predicted sleep disturbance (β = 0.31, t = 12.20, *p* <0.001). In addition, as shown in Table 5, bootstrap 95%CIs for all paths did not include zero, indicating that somatic symptoms played a partial mediating role in the relationship between hostility and sleep disturbance. The mediating effect of somatic symptoms accounted for 51.68% (0.169/0.327 * 100%) of total effects.

### 3.5. Moderating Effect of OCS on the Mediator Effect

As shown in Table 6, somatic symptoms were predicted significantly by hostility (β = 0.52, t = 23.77, *p* < 0.001) and OCS (β = 0.09, t = 3.86, *p* < 0.001), but the interaction item of hostility and OCS (X × W in Table 6) did not predict somatic symptoms significantly. Sleep disturbance was predicted significantly by somatic symptoms (β = 0.26, t = 9.36, *p* < 0.001), OCS (β = 0.38, t = 16.60, *p* < 0.001) and the interaction item of somatic symptoms and OCS (β = 0.01, t = 0.50, *p* < 0.001); however, the interaction item of hostility and OCS did not significantly predict somatic symptoms (β = 0.01, t = 0.50, *p* = 0.621) and sleep disturbance (β = 0.01, t = 0.17, *p* = 0.867). The moderated mediation model diagram is displayed in Figure 2.

The moderating effect of OCS on the relationship between somatic symptoms and sleep disturbance is displayed in Figure 3. A simple slope test showed that the trend of sleep disturbance score increased significantly as somatic symptoms increased in the condition of low OCS (β = 0.1, t = 4.22, *p* < 0.001) and high OCS (β = 0.16, t = 9.57, *p* < 0.001), as shown in Table 5. Specifically, as somatic symptoms increased by 1 standard deviation, the score of sleep disturbance increased by 0.1 standard deviation (low OCS) and 0.16 standard deviation (high OCS), and a significant difference in the indirect effects was observed between low OCS and high OCS (β = 0.06, t = 2.27, *p* < 0.05).

## 4. Discussion

The main purpose of this study was to explore the relationship between internalizing/externalizing psychological problems and sleep disturbance, and to examine the hypothesized model of how and when crucial internalizing/externalizing psychological problem indicators influence sleep disturbance. To the best of our knowledge, this is the first comprehensive study exploring crucial predictors and mechanisms of sleep disturbance in Chinese college students with mental health problems. Our results showed that some internalizing problems indicators (somatic symptoms) and externalizing problems indicators (hostility and obsessive-compulsive symptoms) are significantly associated with sleep disturbance. Furthermore, our results generally support the hypothesized indirect path between hostility and sleep disturbance. Specifically, our results showed that somatic symptoms exerted a mediating effect on the relationship between hostility and sleep disturbance. In addition, our results also showed that obsessive-compulsive symptoms played a moderating role in the mediating path between somatic symptoms and sleep disturbance. 

Unlike previous studies demonstrating associations between sleep and mental health/distress (anxiety and depression) [1,4,7,12,32], the present study found that sleep disturbance is significantly predicted by hostility, somatic symptoms and OCS rather than anxiety and depression. Discrepancies in crucial predictors of sleep disturbance may be a result of several different factors, including differences in the measurement and participants. We used comprehensive measures to assess internalizing mental problems (seven indicators, including anxiety and depression) and externalizing mental problems (seven indicators), which provided us with various indicators to assess which are significant risk factors for sleep disturbance. In comparison with normal college students, college students with various mental health problems are an at-risk group that needs effective interventions to reduce possible hazards. Associations between mental health problems and sleep disturbance are different for the two groups. Our findings that hostility, somatic symptoms and OCS are significantly correlated with sleep disturbance are in line with previous evidence in the recent literature. For example, Tsuchiyama et al. found that increased hostility is significantly correlated with poor sleep quality [14]. In addition, Kundu et al. suggested that somatic symptoms have bidirectional relationships with sleep disorders [18]. Li et al. indicated that obsessive-compulsive disorder is associated with insomnia among patients with depression [16].

Furthermore, our results showed that the associations between hostility and sleep disturbance were mediated by somatic symptoms. Hostility predicted somatic symptoms significantly, which subsequently exerted an impact on sleep disturbance. The role of hostility in somatic symptoms has been verified in previous studies, which found that the level of hostility was significantly correlated with the severity of somatic symptoms involved in the neuromuscular system and the gastrointestinal organ system [28,33], and Nordin et al. suggested that it appeared to be particularly common in sleep disturbance to have gastrointestinal symptoms such as stomach pain, nausea/gas/indigestion and constipation/loose bowels/diarrhea [13]. Of concern, once obsessive-compulsive symptoms were added to the mediating model as a moderator, sleep disturbance was significantly predicted by somatic symptoms, OCS and the interaction item of somatic symptoms and OCS rather than hostility or the interaction item of hostility and OCS. These results indicated that there are interaction effects between somatic symptoms and OCS on sleep disturbance. Moreover, the results showed a significant difference in the mediating effects on sleep disturbance between low OCS and high OCS. These results can be further explained by a high level of OCS exacerbating the adverse effects of somatic symptoms on sleep disturbance in this population. Overall, the results support our hypothesized moderated mediation model.

However, several limitations should be mentioned. First, the data was collected from a self-reported and retrospective survey, from which the information obtained tends to be subjective. Therefore, future research should consider an integrative approach that includes objective measures, such as actigraphy for assessing sleep disturbance. Second, the sample in the present study was only from one large university in Shandong province in China, which may reduce the generalizability of the findings. Third, the cross-sectional study design means we should be cautious about inferring cause-effect relationships. Future studies may elect to use longitudinal or interventional study designs to explore the potential causal relationships between internalizing/externalizing problems and sleep disturbance.

Despite these limitations, the current study benefits from several strengths. It is the first study to use a wide range of indicators related to mental health problems, including hallucinations and delusions, suicidal thoughts, school adaptation, academic stress and interpersonal or intimate relationship difficulties. Thus, our study can assess the unique associations of various mental health symptoms with sleep disturbance more thoroughly. Moreover, our main findings were that the mediating effect of somatic symptoms on the relationship between hostility and sleep disturbance was moderated by OCS. In comparison with the low OCS group, somatic symptoms in the high OCS group exert more adverse impacts on sleep. These findings have important implications for designing practical intervention strategies. First, college mental health workers can assess markers of student hostility, including anger, aggression and cynicism, and then work to reduce their hostility, for hostility is closely related to somatic symptoms that can influence sleep disturbance significantly. Second, given the unique associations between somatic symptoms, OCS and sleep disturbance, college mental health workers should also emphasize the assessment and intervention of somatic symptoms and OCS to improve sleep quality in college students with mental health problems. For instance, they could use cognitive behavioral therapy (CBT) as a treatment to help students focus on present, reasonable thoughts and beliefs about physical conditions and surroundings. In sum, apart from considering sleep disturbance intervention such as cultivating sleep hygiene habits, the evaluation of hostility, somatic symptoms and OCS should also be conducted, and improvement in hostility, somatic symptoms and OCS is expected to improve sleep in college students with mental health problems.

## 5. Conclusions

The present study is an important first step in exploring the impact of a wide range of internalizing/externalizing psychological problems on sleep disturbance. Our work examined a large sample of Chinese college students with mental health problems and the results verified our hypotheses. Specifically, hostility, somatic symptoms and OCS are positively associated with sleep disturbance, and somatic symptoms played a mediating role in the relationship between hostility and sleep disturbance. Furthermore, the mediating role was moderated by OCS, and the mediating effects of somatic symptoms on sleep disturbance were significantly different between low OCS and high OCS. High levels of OCS exacerbated the adverse effects of somatic symptoms on sleep disturbance. Together, these key findings can be described by a moderated mediation model of sleep disturbance in college students with mental health problems.

## Figures and Tables

**Figure 1 ijerph-19-14570-f001:**
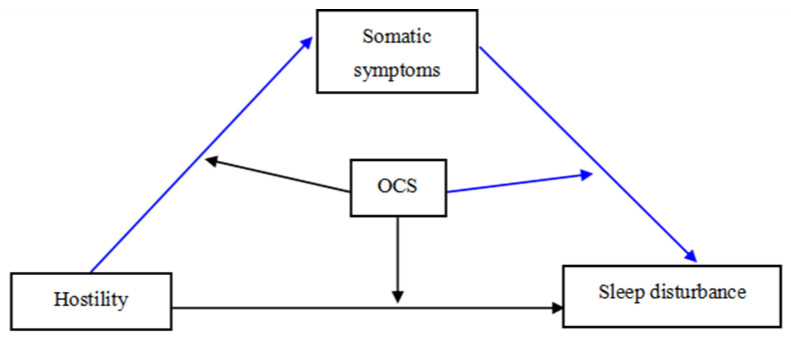
Diagram of the theoretical moderated mediation model.

**Figure 2 ijerph-19-14570-f002:**
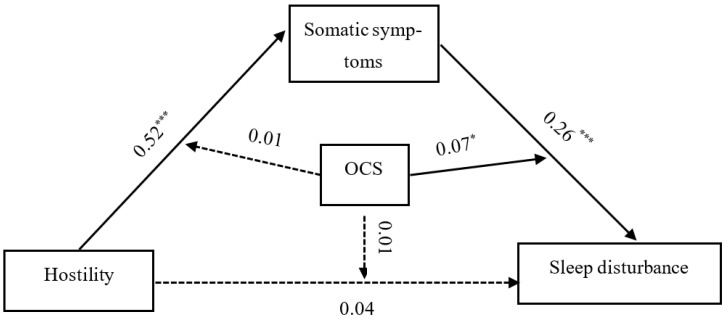
Diagram of the moderated mediation model. Note: Significant pathways are represented by solid arrows, and nonsignificant pathways are represented by dotted lines. Note: * means *p* < 0.05; ***** means p <0.001.

**Figure 3 ijerph-19-14570-f003:**
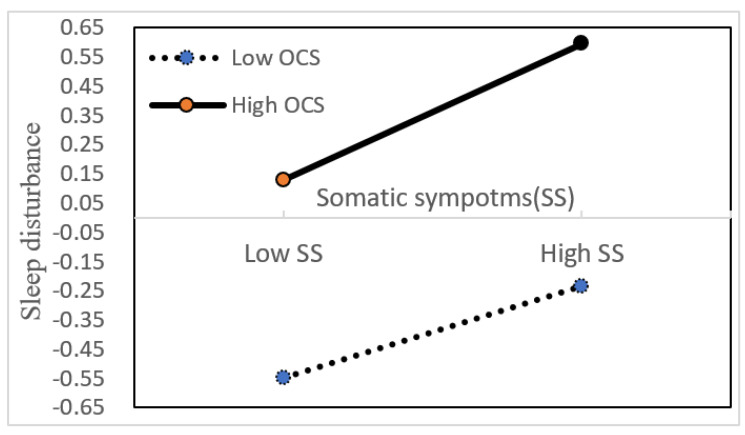
The moderating effect on relationship between somatic symptoms (SS) and sleep disturbance. Note: All variables in the model have received standardized treatment.

**Table 1 ijerph-19-14570-t001:** Chinese college students’ mental health screening scale and criteria.

Level	Category	Indicators	Items	Internal Consistency Reliability	Criteria for Screening Participants
I: Severe Psychological problems		Hallucinations and delusions	4	0.76	Either standard score (Z_I_) > 1
		Suicidal thoughts	4	0.86	
II: General psychological problems	Internalizing problems	Anxiety	4	0.79	Any indicator standard score (Z_II_) > 2
		Depression	5	0.81	
		Paranoia	4	0.82	
		Inferiority	5	0.86	
		Sensitivity	4	0.80	
		Social phobia	4	0.82	
		Somatic symptoms	4	0.78	
	Externalizing problems	Dependence	4	0.83	
		Hostility	4	0.77	
		Impulsivity	4	0.79	
		Obsessive-compulsive symptoms	4	0.78	
		Internet addiction	5	0.88	
		Self-injurious behavior	4	0.78	
		Eating problems	4	0.59	
III: Developmental disturbance problems or psychological problems source		School adaptation	4	0.68	Any indicator standard score (Z_III_) > 3
		Interpersonal relationship	4	0.82	
		Academic stress	4	0.82	
		Employment stress	4	0.88	
		Intimate relationships troubles	4	0.65	
Sleep disturbance			4	0.72	Standard score (Z_sd_) > 2

**Table 2 ijerph-19-14570-t002:** Prevalence and comparison of sleep disturbance stratified by different sociodemographic characteristics of college students with mental health problems (*n* (%)).

Variables	Total	Sleep Disturbance No. (Rate, *%*)	*Permutation p*-Value
**Number**	2134	78 (3.7)	
**Sex**			0.399
Men	1285 (60.2)	43 (55.1)	
Women	849 (39.8)	35 (44.9)	
**Only child**			0.117
No	1357 (63.6)	43 (55.1)	
Yes	777 (36.4)	35 (44.9)	
**Hometown**			0.102
Large cities	202 (9.5)	11 (14.1)	
Small- and medium-sized cities	506 (23.7)	21 (26.9)	
Small towns	356 (16.7)	18 (23.1)	
Rural areas	1070 (50.1)	28 (35.9)	
**COVID-19**			0.487
Before	997 (46.7)	33 (42.3)	
During	1137 (53.3)	45 (57.7)	

**Table 3 ijerph-19-14570-t003:** Internalizing/externalizing problem indicators ($\bar{x}$ ± *s*) associated with sleep disturbance examined by multivariate binary logistic regression analyses (*n* = 2134).

Variables	No Sleep Disturbance (*n* = 2056)	Sleep Disturbance (*n* = 78)	Odds Ratio (95% Confidence Interval)
Hostility	7.25 ± 1.96	9.45 ± 3.53	1.20 (1.06–1.36) ^***^
somatic symptoms	7.05 ± 2.18	9.33 ± 3.38	1.31 (1.17–1.48) ^***^
OCS	8.27 ± 2.19	10.97 ± 3.30	1.32 (1.18–1.48) ^***^

Note. *** *p* < 0.001.

**Table 4 ijerph-19-14570-t004:** The mediating effect test of somatic symptoms.

Outcome Variables	Predictors	*R*-Squared	*p*	*β*	Standard Error	*t*
M: Somatic symptoms	X: Hostility	0.30	<0.001	0.55	0.02	25.88 ^***^
Y: Sleep disturbance	X: Hostility	0.17	<0.001	0.16	0.03	5.85 ^***^
	M: Somatic symptoms			0.31	0.03	12.20 ^***^

Note. *** *p* < 0.001.

**Table 5 ijerph-19-14570-t005:** Total effects, direct effects and mediating effects in the mediation model (bootstrap = 5000).

	Standardized Effect Size	Standard Error	95% Confidence Interval (CI)	Relative Effect Size
Total effects	0.327	0.02	0.28–0.38	
Direct effects	0.158	0.03	0.11–0.21	48.32%
Mediating effects	0.169	0.02	0.14–0.20	51.68%

**Table 6 ijerph-19-14570-t006:** The moderating effect test of OCS on the mediator effect.

Outcome Variables	Predictors	*R*-Squared	*p*	β	Standard Error	*t*	*p*	95% Confidence Interval
M: somatic symptoms	X: Hostility	0.31	<0.001	0.52	0.02	23.77	<0.001	0.47–0.56
	W: OCS			0.09	0.02	3.86	<0.001	0.04–0.13
	X × W			0.01	0.02	0.50	0.621	−0.03–0.04
Y: Sleep disturbance	X: Hostility	0.29	<0.001	0.04	0.03	1.36	0.174	−0.02–0.10
	M: somatic symptoms			0.26	0.03	9.36	<0.001	0.20–0.31
	W: OCS			0.38	0.02	16.60	<0.001	0.33–0.42
	X × W			0.01	0.02	0.17	0.867	−0.04–0.05
	M × W			0.06	0.03	2.26	<0.05	0.01–0.10
Conditional mediating effect on Y at values of the moderator								
	Low OCS			0.10	0.02	4.22	<0.001	0.06–0.15
	Mean OCS			0.13	0.02	8.31	<0.001	0.10–0.16
	High OCS			0.16	0.02	9.57	<0.001	0.13–0.20
	Differences between high and low OCS			0.06	0.03	2.273	<0.05	0.01–0.12

Note: All variables in the model have received standardized treatment.

## Data Availability

The data presented in this study are available from the corresponding author. The data are not publicly available due to privacy and ethical considerations.

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
