# Peer review of "Sleep Disturbance in Chinese College Students with Mental Health Problems: A Moderated Mediation Model"

_ijerph, 2022, doi:10.3390/ijerph192114570_

Round 1

Reviewer 1 Report

The authors of this work presented an interesting study of the relationship between hostility, somatic symptoms and OCS with sleep disturbance using a novel application of a moderating mediating model.

The main comment that I have is that the introduction could have a few sentences about this model and why it is used for a study like this.  Any sort of background information that can be provided to the reader about this model would be helpful for those that are new to the application.

Author Response

Dear Reviewer,

On behalf of my co-authors, I would like to express my gratitude to you for insightful and helpful comments of our manuscript entitled Sleep Disturbance in Chinese College Students with Mental Health Problems: A Moderated Mediation Model” (ijerhp-1923933).

We carefully considered your suggestions, and we made the revisions that are shown in red in the manuscript. We did our best to update our manuscript in response to the comments. Please see the updated manuscript attached and the following responses, which we are submitting for your kind consideration.

Comment:  The main comment that I have is that the introduction could have a few sentences about this model and why it is used for a study like this.  Any sort of background information that can be provided to the reader about this model would be helpful for those that are new to the application.

Response: We greatly appreciate the reviewer’s comments for raising an excellent point. Thus, we add the role of moderated mediation model analyses in exploring the impact mechanism of sleep disturbance. Specifically, the following content has been added into the revised manuscript.

Page 3, Line 104-110. A second objective was to examine how and when these crucial internalizing/externalizing problems exert impacts on sleep disturbance using a moderated mediation model as shown in Figure 1. The moderated mediation model analyses can handle issues such as how a predictor variable impacts an outcome variable and when a variable mostly strongly predicts an outcome variable. Thus, we adopt the mediation model analyses to evaluate quantitatively the extent to which a predictor variable may affect an outcome variable through a potential mechanism [30], and whether the indirect effects of the predictor variable on the outcome variable alters in high/low levels of a moderator.

Page 3, Line 114-116. (hostility   somatic symptoms   sleep disturbance). In other words, hostility has indirect effects on sleep disturbance through somatic symptoms.

Page 3, Line 119-122. The moderator variable OCS is a variable that affects the strength of the relationship between somatic symptoms and sleep disturbance. That is, the mediating effect of somatic symptoms on sleep disturbance depends on the levels of the moderator OCS.

Page 3, Line 124-127. Fourth, we hypothesized that the mediating effect of somatic symptoms on sleep disturbance is significantly higher in high levels of OCS than in low levels of OCS. In the high levels of OCS, somatic symptoms will exert more negative impacts on sleep disturbance.

Page 3, Line128-129. “Figure 1 Diagram of the theoretical moderated mediation model” was added into the revised manuscript.

Figure 1 Diagram of the theoretical moderated mediation model

Best regards.

Yanping Sun

Reviewer 2 Report

the manuscript is descriptive, and it is not appropriate for the journal.

Author Response

Dear Reviewer,

On behalf of my co-authors, I would like to express my gratitude to you for your comments of our manuscript entitled Sleep Disturbance in Chinese College Students with Mental Health Problems: A Moderated Mediation Model” (ijerhp-1923933).

We carefully considered your suggestions, and we made the revisions that are shown in red in the manuscript. We did our best to update our manuscript in response to the comments. Please see the updated manuscript attached and the following responses, which we are submitting for your kind consideration.

Comment:  the manuscript is descriptive, and it is not appropriate for the journal.

Response: Thank you for the comment. We respectfully disagree with the comment, the reasons are as follows:

  • Our study is not a descriptive study, because we adopt a moderated mediation model to examine how hostility affects sleep disturbance and when the moderator variable (obsessive-compulsive symptoms, OCS) mostly strongly predicts sleep disturbance among Chinese college students with mental health problems. We formulate hypotheses on the impact mechanism of sleep disturbance, and we draw inferences about the direct/indirect effects of hostility on sleep disturbance based on the model analyses using Mplus Version 7.4. The study is not a descriptive or qualitative study.
  • Sleep disturbance has become a notable public health concern in college students. For example, the Annual Sleep Report of China 2022 reported that more than 90 percent of Chinese college students had average (49%) and poor (45%) sleep quality. Lund and her colleagues found that about 60% of American college students have sleep disorders, and 17.9% use hypnotic drugs to improve their sleep. Effective sleep is crucial for the development of physical, cognitive and psychosocial functioning, and sleep is a fundamental physiological and psychological process for our mental health and welfare. The study can offer empirical support for the prevention of sleep disturbance in college students by examining crucial predictors and mechanism of sleep disturbance. It belongs to the public health scope of the journal.

In sum, the study conducts moderated mediation model analyses to examine the mechanism of sleep disturbance in Chinese college students, and it is a quantitative study appropriate for the journal.

Best regards.

Yanping Sun

Reviewer 3 Report

The authors investigated the impact of a wide range of internalizing/externalizing psychological problems on sleep disturbance in Chinese college students with mental health problems. This is an interesting and meaningful study. However, I have several minor concerns before the paper could be considered for publication.

1. The language needs to be improved.

2. In table 3, the sample sizes of No sleep disturbance (n=2056) and Sleep disturbance (n=78) have a huge difference, chi-square test maybe invalid. Please use permutation test.

3. Please define the abbreviation in its first appearance, such as OR, CI, R^2, SE,

4. What’s the basis of the model shown in figure 1. Please add it the method part.

Author Response

Dear Reviewer,

On behalf of my co-authors, I would like to express my gratitude to you for insightful and helpful comments of our manuscript entitled Sleep Disturbance in Chinese College Students with Mental Health Problems: A Moderated Mediation Model” (ijerhp-1923933).

We carefully considered your suggestions, and we made the revisions that are shown in red in the manuscript. We did our best to update our manuscript in response to the comments. Please see the responses attached, which we are submitting for your kind consideration.

Best regards.

Yanping Sun

Reviewer 4 Report

There are a few changes/corrections that should be addressed prior to publication.

1. Page 1, Line 42, "...nearly half of the college students went to bed after 12 p.m", authors might need to check whether it is 12am or 12pm? 

2. Suggest to introduce the full term of abbreviation, e.g., OR (page 5)

3. The moderated mediation model presented in this study is a unidirectional mediation model, it would be good if authors consider and examine the bidirectional mediation model to estimate the direct and indirect effects.

Author Response

(The authors gave the same response as above.)
